# Ore Minerals and Metal Distribution in Tailings of Sediment-Hosted Stratiform Copper Deposits from Poland and Kazakhstan

**Agata Duczmal-Czernikiewicz** [1,*] **, Adilkhan Baibatsha** [2] **, Alma Bekbotayeva** [2] **, Gulnara Omarova** [2] **and Akmaral Baisalova** [2]

1. Mineralogical Research Laboratory, Faculty of Geography and Geological Sciences, Institute of Geology, Adam Mickiewicz University in Poznan, 61-680 Poznan, Poland
2. Department of Geological Survey, Prospecting and Exploration of Mineral Deposits, Institute of Geology, Oil and Mining, Satbayev University (Kazakh National Research Technical University named after K.I.Satbayev), Almaty 050013, Kazakhstan; baibatsha48@mail.ru (A.B.); a.bekbotaeva@mail.ru (A.B.); omarova.gulnara@mail.ru (G.O.); baisalova73@mail.ru (A.B.)
* Correspondence: agata.duczmal-czernikiewicz@amu.edu.pl

**Abstract:** This study, carried out in tailings from two sediment-hosted stratiform copper deposits in the Lublin-Głogów Copper District in Poland (Kupferschiefer-type deposit) and Zhezkazgan (cupriferous sandstone-type deposit) in Kazakhstan, analysed the mineralogy of copper, zinc, and lead minerals as related to metal accumulation in sediments. Microscopic study in reflected light and SEM–EDS (Scanning Electron Microscope—Energy Dispersive Spectrometer) analysis, as well as chemical diversity in the used INAA (Instrumental Neutron Activation Analysis), ICP (Inductively Coupled Plasma), and AAS (Atomic Absorption Spectroscopy) methods in 35 samples from Kazakhstan and 35 from Poland were examined due to their diversity. In both tailing deposits in Kazakhstan and Poland, heavy fractions were dominated by copper sulphides: chalcopyrite ($CuFeS_2$), bornite ($Cu_5FeS_4$). and chalcocite ($Cu_2S$). Moreover, sphalerite, galena, and cerussite have been recognized as a carriers of Zn and Pb. Their geochemistry was dominated by Cu, showing a mean content of 2500 ppm, in both Poland and Kazakhstan. Zinc and lead also occurred, showing a content of approximately 200 ppm and 500 ppm in Poland, and 1500 ppm Zn and 2500 ppm Pb in Kazakhstan, respectively. Grain size analysis indicated that the dominant grain size in both districts corresponded to the silt and fine sands fractions. Copper, zinc and lead sulphides accumulated in fine fractions in tailings from Kazakhstan (in sandstones and quartz grains), and mainly in coarse fractions in Poland (within carbonates, sandstones, and black shales). Mineralogical and geochemical features should be taken into consideration when assessing potential metal sources of technogenic materials.

**Keywords:** sediment-hosted stratiform copper deposits; tailings; sulphides; mineralogy; base metals; Poland; Kazakhstan

## 1. Introduction

The exhaustion of natural resources is a global challenge in the 21st century [1–4]. A large portion of the already-discovered deposits of natural resources are in the later stages of exploitation. Despite the discovery of new deposits of various genetic types, many sediment-hosted stratiform copper (SSC) deposits constitute a considerable source of copper and are the subject of a search for new areas of metallic ore extraction [5–7]. One of the additional sources of development in the resource base of currently functioning enterprises is tailings from processing plants, which can be considered as technogenic deposits. This new type of mineral resource can help in the efficient functioning of the existing mining enterprises, as well as the expansion and prolongation of their activity. Recovery and recycling of waste also corresponds with the global trend of the functioning of economies based on sustainable development [8]. Until recently, Kazakhstan was one of

the top ten countries in the world in terms of copper production [9]. With the exhaustion of traditional ores in the Zhezkazgan deposits, however, the volume of extracted copper has considerably decreased. Poland is in the top 20 countries producing copper, and among the global leaders in terms of silver production [9]. Copper ore resources in Lubin, such as the Głogów Copper District (Legnicko–Głogowski Okreg Miedzionośny (LGOM)), are continuously declining, although new deposits have been recently discovered north and north-east of the currently exploited ones [10].

Deposits of the SSC type, and porphyry copper deposits, constitute the most important copper sources in the world [5,11,12]. Approximately 20% of global copper comes from this genetic type of deposit. SSC deposits are the economic pillars of many countries, including Poland, the Democratic Republic of Congo, and Kazakhstan [11,12]. In Kazakhstan, until recently, copper deposits of this type exceeded 70% of all copper resources, at an annual level of production of more than 580 million tonnes [6,11]. Poland is 12th in the world in terms of copper production (400 million tonnes of copper annually). Next to copper, SSC deposits are a valuable source of silver, gold, and platinum group elements. Moreover, SSC deposits are an important global source of cobalt [5,13–15]. A common feature of SSC deposits is their relationship with sedimentary rocks, often in the form of deposits with a relatively small thickness and high range. The thickness of Polish deposits ranges from 0.5 to 10 m [10], and in Kazakhstan, the total thickness of the deposits was 40 m [16]. The Zhezkazgan deposit is unique in Kazakhstan in terms of the number of mineralised horizons and compound mineral composition of ores [16,17]. With the exhaustion of ore resources, however, the volume of produced copper considerably decreased (about 230,000 tonnes in 2015) [18].

Mining wastes, collected in the process of sedimentation after ore floatation, are an integral part of the copper industry. They are an inseparable part of metal extraction, although they usually constitute useless remains from ore processing. Tailing ponds collect the resources of valuable components that, according to Polish law, cannot be recovered. On the other hand, waste deposits can be a threat to the environment, not only due to the content of hazardous components, such as arsenic and cadmium [19,20], but also due to the possibility of damaging dams and retaining tailing ponds in the safety zone [21,22]. Although, in Poland, metal recovery from tailing ponds is not possible, the National Programme of Geological Research for the years 2021–2025, adopted in Kazakhstan, permits research and undertakes work on the processing of post-floatation waste.

In Poland, tailing ponds and the accumulated waste are managed by the Polish State of Mining and Metallurgical Combine (Polish Copper District), and were numerously analysed in terms of the content of both useful and hazardous components [19,23,24]. In Kazakhstan, the material has not been subject to extensive research, either geological or technological. The tailing deposits of Zhezkazgan have accumulated more than 1 billion tonnes of waste [25], whereas Polish tailing ponds contain 600 million tonnes, with an approximate copper content between 0.1% and 0.3% [26]. The total geological resources of metals accumulated in tailing ponds may amount to at least 3.0 million tonnes in the Zhezkazgan region. Based on these estimates, the tailings can be considered promising and useful in the processing of technogenic mineral resources. Research into the mineral composition and distribution of metals in tailings can contribute to the development of technological research regarding the extraction of metals from materials of anthropogenic origin, which can be treated as potential deposits of copper and other primary and noble metals [20,27]. Tailings can be described as technogenic deposits and can be more or less interesting for the local economies, depending on the availability of metals in the global economy, the existing law, and the resource policies of respective countries [28–32].

The subject of the study was copper ore floatation waste from dams of three tailing ponds: Żelazny Most (ZM) in Poland, and two tailing ponds in Kazakhstan: Borgezsai (B) and Osnovnoe (OS). The mineral composition and concentrations of primary and trace components were determined separately, by means of the same or comparable methods of analysis. Concentrations of copper, zinc and lead, as well as the accompanying metals, were

determined in reference to the characteristics of the mineral components of the deposits. Petrographic properties of the deposits in tailing ponds were described in reference to the characteristic properties of SSC deposits. The calculated statistical parameters pointed to the high similarity of tailing deposits in two large SSC deposits in Poland and Kazakhstan.

## 2. Copper Deposits in Poland and Kazakhstan

### 2.1. Sedimentary-Hosted Stratiform Copper Deposits (SSC) in Poland, Lower Silesia

Copper deposits in Lower Silesia (Figure 1) have been exploited in the Old Copper Belt in Sudety since the 9th century. Since 1957, it has been exploited on the Fore-Sudetic Monocline in the vicinity of Lubin and Głogów [33]. Several mining areas occur there: Lubin-Małomice and Rudna, Sieroszowice-Polkowice, Radwanice, Gaworzyce, Głogów Głęboki [34,35], and, since 2019, the Żory deposit [10]. The area of copper deposits has a length of 60 km and width of 20 km. Copper ores are currently exploited in the Lubin, Polkowice-Sieroszowice, and Rudna mines from minerals containing copper, which occur in copper-bearing shale, sandstone, dolomite, and limestone. The richest mineralisation occurs in black copper-bearing shales, described by the German name Kupferschiefer [7,36]. Copper ores accumulate in deposits, forming sedimentary-hosted stratiform copper deposits (SSC) of the Kupferschiefer type [37].

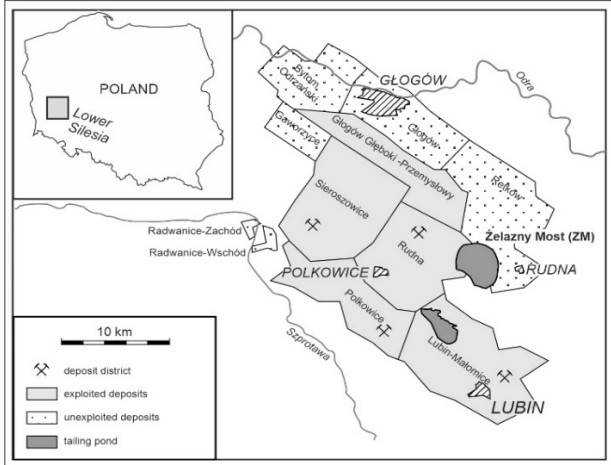

**Figure 1.** Lubin-Głogów Copper District (Lower Silesia) with copper deposits and the location of tailings.

The ores are located at a depth ranging from 300 m in the southern part to 1500 m in the northern part. The primary technological ores are made of sandstones (bottom part), black shales, and carbonates. Copper-bearing shale (Kupferschiefer) is the richest part of the deposit in terms of metal content. It usually constitutes the boundary between the rocks of red sandstone (Lower Permian) and Zechstein (Upper Permian) [36–38]. The primary copper minerals occurring in Polish deposits include chalcocite, bornite, and chalcopyrite, as well as other copper, silver, lead, zinc, and cobalt minerals. In 2019, economic copper ore resources were estimated at up to 1951.2 million tonnes (including 34.8 Mt of metallic copper and 103.6 k t of silver). In 2019, mean copper content was 1.50%, and silver content was 48.69 g/t. As a by-product, metals such as Ag, Au, Ni, Pb, Se, and Re are recovered from copper ores. Co, Mo, Bi, As, Sb, Hg, Pt, Pd, V, U, and Th occur in lower amounts. In 2019, 1400 t Ag, 674 kg Au, 28.5 kt Pb, 2 kt $NiSO_4$, 75.8 t Se, and 8.3 t Re were produced from the copper ores exploited in Poland. The total forecasted resources of copper and silver ores, calculated up to a depth of 2000 m, are 10.3 Mt, prospectively, they are 15.7 Mt, and hypothetically, they are 8.8 Mt Cu [38].

## 2.2. Sedimentary-Hosted Stratiform Copper Deposits (SSC) in Zhezkazgan, Kazakhstan

The mineralisation of the Zhezkazgan deposit focuses on two ore-bearing formations—Taskudyk and Zhezkazgan—with 10 identified ore horizons composed of 44 ore bodies [16,18,39]. The primary ore minerals of the deposit include bornite, Cu-Fe minerals from the chalcocite group, chalcopyrite, galena, and sphalerite. As in the case of other SSC deposits, the primary accompanying components with industrial importance in deposits include Cu, Zn, and Pb [39,40]: silver, gold, rhenium, osmium, cadmium and, in smaller amounts, cobalt, nickel, molybdenum, tin, bismuth, arsenic, antimony, tellurium, selenium, gallium, indium, thallium, germanium, palladium, platinum, and mercury. No exploitation is currently conducted for these deposits.

## 3. Tailings

### 3.1. Tailings in Poland

Metal deposits are recovered from the exploited minerals in the processes of ore dressing, which cover crushing, grinding, separation (floatation), and metallurgic processes. The processing results in a copper concentrate with a copper content ranging from 18 to 32% (on average, 24%) [24,26]. The pulp obtained from floatation constitutes a concentrate of copper, zinc, and lead. Silver and other fine metals are recovered over the course of additional floatation stages. After separation of the concentrate, water suspension of tailing remains, with a Cu content of 0.17–0.27% [41]. The remains are transported through a system of pipelines to the tailing pond. In Poland, the tailing under study is located in the LGOM area and is called the Waste Management Facility. It is the largest tailing pond in Europe, and one of the largest in the world. The Żelazny Most (ZM) facility has collected 660 million m$^3$ of tailings to date. The expansion of the facility is currently underway, increasing its surface area to approximately 2100 ha. After the expansion, it will have the capacity to store up to 950 million m$^3$ of waste.

### 3.2. Tailings in Kazakhstan

The tailing ponds Borgezsai (B) and Osnovnoe (OS) in the Zhezkazgan collect vast resources of technogenic ores, which offer a measurable perspective of the expansion of the base of mineral resources and can be processed for the production of copper and other metals [42,43]. The Borgezsai (B) tailing deposit stores' waste derives from ore processing, with an amount of approximately 48,990 tonnes, and from a copper content of 0.1–0.3%. The copper resources contained in the deposit are estimated at approximately 93,108 t, with a mean Cu content of 0.19% and a lead content of 4465 t, with a Pb content of 0.02–0.15%, indicating more than 1.6 billion tonnes of technogenic ore resources. Based on the estimates of Kazakhstan LLP, over the last 45 years (from 1964 to 2008), the Osnovnoe (OS) tailing pond has accumulated 852,813 thousand tonnes, or 582,522,886 m$^3$, of waste from the Zhezkazgan ore dressing plant. Metals are recovered from exploited minerals in the processes of ore dressing, including crushing, grinding, separation (floatation), and metallurgic processes. The amount of copper is 1,094,502 t Cu (with mean Cu content of 0.128%), Pb = 170,419 t (0.02%), Zn = 253,580 t (0.03%), and Ag = 2,098,833 t (2.46 g/t). In the new tailing deposit, where tailings from the processing plant have been deposited since 2008, more than 15,000 million tonnes of technogenic sediments have been collected to date.

## 4. Materials and Methods

### 4.1. Sampling and Sample Preparation

Deposits in ZM were sampled from boreholes with lengths of 1.5 and 5 m, performed by means of a manual core and from the surface of the tailing dams (Figure 2A–D). A total of 35 samples were collected, every 0.5 m. The collected samples were sieved on a set of sieves, and the heavy fraction was separated for the preparation of microscope slides. On the selected samples, thin sections were polished for microscopic observations in reflected light, as well as for research in the micro-area using SEM–EDS. The samples were not

coated before the SEM–EDS analysis. Samples not separated into fractions were analysed by means of the INNA and ICP-MC method.

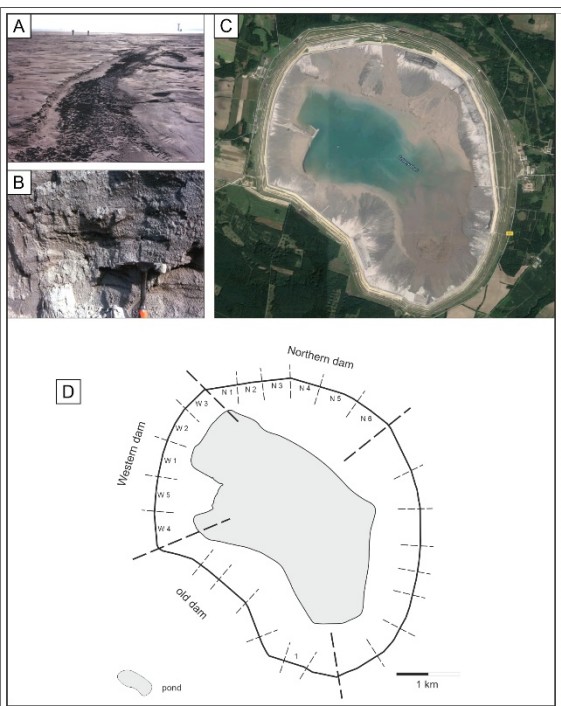

**Figure 2.** (**A**) Photograph of the ZM northern dam, view from the crown; (**B**) enlargement of A; sedimentary record in a profile, hammer as a scale; (**C**) ZM tailing location on Google Maps (accessed 27 May 2019); (**D**) sampling scheme from northern and western dam, the old dam on the picture before reconstruction (before 2021).

In the Zhezgazgan tailings, samples were collected from boreholes and from the surface of the tailing storage. Depending on the depth of the Borgezsai tailing deposit, boreholes were drilled with depths from 4.7 to 13.8 m. (Figure 3). A total of 14 samples were collected from the core, and 17 from the surface of the tailing pond. A total of 31 samples were analysed. According to borehole data, the depth of the tailing deposit is approximately 23.0 m. A total of 35 samples were collected from cores from the Osnovnoe tailing deposit, with a depth of approximately 57 m. The locations of boreholes were selected based on technical coring feasibility, and the length of the borehole was based on the financial feasibility of the project. The length of the sampled section, depending on the sampling conditions, ranged from 0.5 to 1.5 m. The selected samples were ground and dried, and those used for spectral and chemical analysis were ground again. Polished, thin sections were used for microscope observations. Samples for laboratory chemical research were collected from the Osnovnoe (OS) tailing pond. The sampling location is presented in (Figure 3A–C). To prepare the samples, they were washed on a sieve with a pore diameter of 0.063 mm with distilled water. The grain size composition of the deposit was analysed by means of the sieve method, by sieving in a sieve column: 2; 1; 0.5; 0.25; 0.16; 0.1; 0.071; and 0.056 mm at the laboratory of the Institute of Geology of the Adam Mickiewicz University. At the laboratory of the Satbaeva University, the set is composed of the following sieves: 4; 2; 1; 0.5; 0.25; 0.125; 0.63; and 0.45 mm.

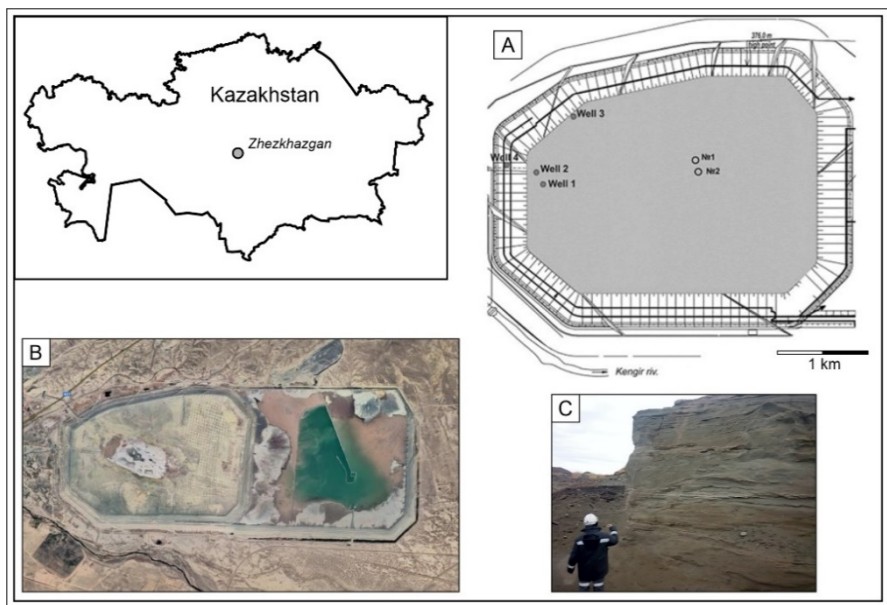

**Figure 3.** Zhezkazgan location, (**A**) drill hole location and sampling scheme in Osnovnoe (OS) tailing, (**B**) Osnovnoe (OS) location, according to Google Maps (accessed 27 May 2019), (**C**) photograph of sampling of the Osnovnoe (OS) indicating interbedded fine silts and coarse sands.

### 4.2. Mineralogy

Optical microscopes were used for the observation of transmitted and reflected light, as well as a scanning microscope, SEM–EDS. The microscope Axioplan (JENAPOL, Zeiss) was used at the Institute of Geology, and SEM–EDS was used at the Scanning Microscopy Laboratory of Geography and Geological Sciences of the Adam Mickiewicz University in Poznań. The polished thin sections were observed in the low vacuum and were not covered by the metal. Mineralogical research was separately conducted at the Laboratory of the Institute of Geology of UAM (35 samples) and the Satbayeva University (35 samples), after collecting samples from the surface material in tailing dams and boreholes from ZM in LGOM and OS in Zhezkazgan (Figures 2 and 3). Quantitative calculations were performed under a microscope with a conventional eyepiece, along with determination of the optical properties of minerals. To avoid repeated counts in one field of vision, the movement of the thin section was required. The essence of the calculation is to assess the part of the field of view that is occupied by a given mineral. After calculating each mineral over the area of the entire thin section, the quantitative ratios of the minerals in it were calculated. The cross of the eyepiece divides the field of view into four equal parts. The percentage can be calculated relative to the entire field of view, given that every quarter is 25%. For example, if a mineral occupies approximately half a quarter of the field of view, then its content in this case amounts to 12.5%, and the fourth part is 6%. The quantitative ratios obtained in one field of view were recorded in a similar way: by moving the thin section, the content of minerals in the remaining parts was calculated over the entire section area. The arithmetical mean was calculated using these data.

### 4.3. Geochemistry

The chemical composition was determined in 35 samples selected from deposits of western and northern dams in the tailing deposits of ZM. The concentrations of components were determined at the Actlab laboratory in Canada (Ontario) by the following methods: INAA (instrumental neutron activation analysis) for the trace elements, As, Ba, Co, Eu, La, Ce, Nd, Sm, Yb, and Lu; and TD-ICP (inductive coupled plasma) for the primary elements, Cu, Cd, Pb, Ni, Zn, Sr, V, and Y. The results of the samples selected for analysis are presented in Tables S1 and S2 (Supplementary Materials). Samples from Kazakhstan were analysed by means of an atomic absorption spectrometer MGA-915MD with a software package

(Russia). Based on geochemical data, the values of statistical parameters were calculated: concentrations of metals, mean, median, and correlation coefficient values in samples from Poland and Kazakhstan. Mean values, medians, and Pearson' correlation coefficients were calculated for the primary and accompanying elements.

## 5. Results

### 5.1. Mineralogical Study

5.1.1. Mineralogical Characteristics of Technogenic Sediments of ZM

ZM deposits are dominated by fragments of ores, primarily sandstone and dolomite, with a smaller share of copper-bearing shales. The heavy fraction reaches 4.8% of the total weight in the western dam and 8% of the total weight in the northern dam. A small portion constitutes ore minerals, which are dominated by iron sulphides: pyrite or marcasite. Single grains of covellite, digenite, chalcocite, bornite, and tennantite seldom occur (Figure 4a). Mineral materials usually occur in the form of intercalations of pyrite with chalcopyrite and bornite with covellite (Figure 4c–f. Copper minerals are very rarely represented by carbonates (malachite). Galena was also recorded (usually together with framboidal pyrite), as well as sphalerite. Titanium oxides are a considerable group, including anatase, rutile, leucoxene, and brookite, as well as the titanium and iron oxides: ilmenite. Moreover, trace occurrences of the spinel group are observed (magnetite and chromite), as well as iron oxides and hydroxides (hematite and goethite), zirconium, tourmaline, apatite, and monazite. The remaining portion is baryte (Figure 4g,h), strontianite, and celestobarite.

Chalcopyrite $CuFeS_2$ occurs in each grain fraction in the form of separate grains, and as a cement in sandstone ores. It has a typical yellow colour in reflected light, weak anisotropy, low VHN hardness, and primarily forms xenomorphic grains. It creates intergrowths with bornite and galena, with covellite, or provides a background for pyrite framboids (Figure 4h). It is characterised by small admixtures of silver. Its content in the heavy fraction does not exceed 3% of the total weight. The chalcopyrite surfaces of this mineral, however, show no signs of weathering. Bornite $Cu_5FeS_4$ occurs with four colour variations: pink, orange, heather, and grey. In reflected light, it is isotropic or weakly anisotropic. It occurs in the form of intercalations with covellite or galena; often, together with covellite, it constitutes a background for pyrite framboids. The intergrowth of bornite with digenite or Fe sulphides is commonly observed. Covellite $CuS$ commonly occurs in the form of separate grains. It also forms intergrowths with bornite, pyrite, or marcasite, digenite, and chalcocite. In the deepest profiles, covellite grains are surrounded by malachite. Digenite $Cu_9S_8$ has optical properties similar to those of chalcocite, whereas it has a darker colour and isotropy. Intercalations of digenite with other $CuS-Cu_2S$ minerals occur very frequently. It also occurs in the form of single grains, and locally with submicroscopic inclusions of galena or framboidal pyrite.

Among copper sulphides, other minerals from the $CuS-Cu_2S$ group were observed: anilite, haykcokite, and djurleite, which are extremely rare in reflected light. These were confirmed by microchemical analyses.

Chalcocite $CuS_2$ occurs in trace amounts. It occurs in the form of separate grains, but can also occur in the form of intercalations with digenite and chalcopyrite, and sometimes with bornite.

Pyrite $FeS_2$ takes the form of framboids or automorphic crystals. Transitions from simple framboids with a size of several microns to polyframboids (with a diameter of up to 50 microns) are observed, sometimes enclosed in automorphic crystals (Figure 4h). Pyrite occurs as a separate mineral ore, or in the presence of titanium oxides, galena, chalcopyrite, sphalerite, or bornite. Apart from copper and lead, it contains no considerable admixtures of other metals (trace amounts of Co). The presence of pyrite was recorded to the amount of up to 50% of the total weight in each of the analysed fractions of heavy grains in the deposits from Żelazny Most. Marcasite ($FeS_2$–rhombohedral) occurs in smaller amounts than pyrite, but frequently co-occurs with pyrite and copper sulphates (covellite, bornite, chalcopyrite), as well as zinc sulphate (sphalerite). It can replace Ti oxides in crystalline

networks of detritic minerals: rutile and leucoxene. It forms concentric structures, as well as rhombohedral automorphic crystals (Figure 4). Galena PbS occurs in the form of intercalations with pyrite, sphalerite, bornite, or chalcocite. It often forms submicroscopic inclusions in sulphate minerals, or is finely dispersed in carbonates, independent clusters and inclusions. It sometimes replaces organic remains in combination with sphalerite (Figure 4g).

On the surface of grains, transformations into minerals from the oxides and hydroxides group, or transformations into carbonate minerals, are observed. Sphalerite ZnS usually accompanies iron sulphates (pyrite or marcasite), copper sulphates (chalcopyrite), and Ti oxides. It commonly occurs in carbonate lithoclasts and does not occur in sandstone fragments. Malachite $Cu(OH)_2CO_3$ forms intercalations with the copper sulphates chalcocite and covellite on their external edges. Malachite commonly forms along the cleavages and is only present in trace amounts in surface samples from ZM. Cerussite $PbCO_3$ is very common mineral, but occurs in small amounts, and always in a close spatial relationship with galena, on its external edges and along its cleavages and cracks.

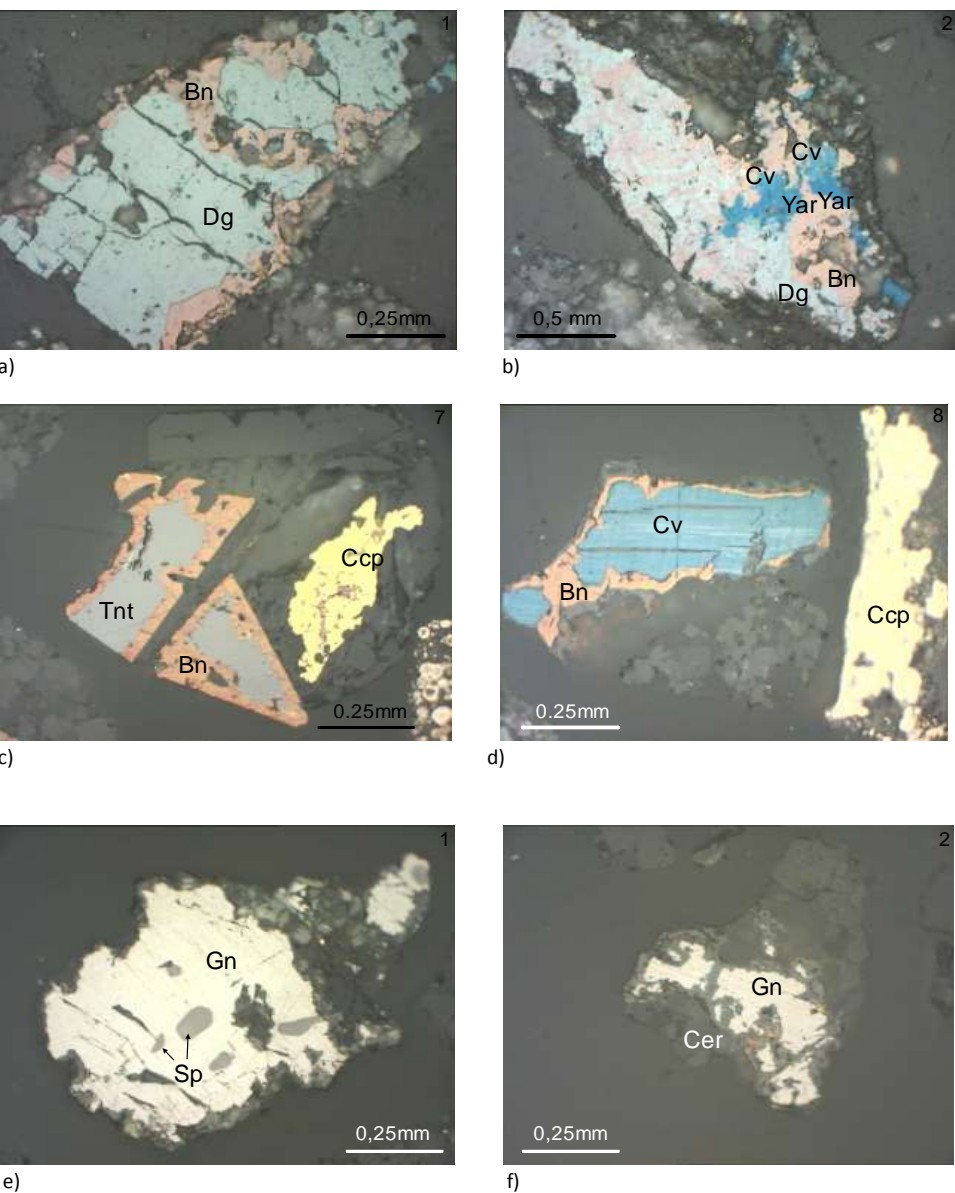

**Figure 4.** *Cont.*

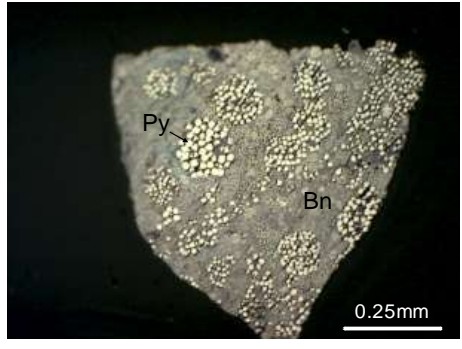
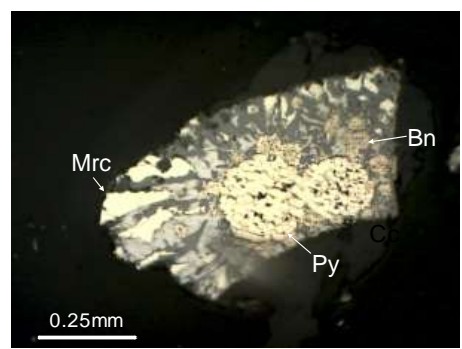

g)                                                          h)

**Figure 4.** Microphotography of ore minerals separated as heavy fractions from ZM sediments under reflected light: (**a**) digenite (Dg) and bornite (Bn) in intergrowths with dolomitic ore ZM, northern dam, N19A; (**b**) bornite (Bn), digenite (Dg), covellite (Cv), and yarrovite (Yar) in sandstone fragment, profile N19A; (**c**) bornite (Bn) overgrowths on tennantite (Tnt) and chalcopyrite (Ccp) included in gangue minerals, profile N19B; (**d**) bornite (Bn) overgrowths on covellite (Cv) as a part of the dolomitic fragment, chalcopyrite (Ccp) as a separate grain, northern dam, profile N19B; (**e**) galena (Gn) and sphalerite (Sp) intergrowths, northern dam, profile N19B; (**f**) galena (Gn) surrounded by cerussite (Cer) western dam, sample W3; (**g**) pyrite framboids cemented by bornite (Bn) western dam, W1; (**h**) pyrite framboids in early form of automorphic grains inside bornite (Bn) and marcasite (Mrc) intergrowths, N19A; 1 polarizer, reflected light.

### 5.1.2. Mineralogical Characteristics of Technogenic Sediments of OS

Microscope analyses of samples of tailings from B and OS showed that copper minerals are dominated by chalcopyrite, followed by covellite, and, more seldomly, chalcocite, bornite, sphalerite, pyrite, and arsenopyrite. The primary size of sulphate grains is 0.01–0.03 mm. Sulphates are primarily located on quartz or are intergrown with it. Intercalations with covellite and chalcopyrite are rarely observed. Galena, sphalerite, and trace amounts of native gold are also present. Chalcopyrite occurs most frequently, constituting from 30 to 50% of samples; the content of covellite ranges from 5 to 20%. Chalcocite content ranges from several to 20% and bornite content ranges from 4 to 25%. Other minerals include galena, sphalerite, rarely pyrite, hematite, magnetite, iron hydroxides, arsenopyrite, rutile, and native gold. Chalcopyrite is the most common copper mineral, with the number of grains ranging from 10 to 95. It rarely occurs in free-form, with a size of 0.01 to 0.07 mm, sometimes reaching up to 0.1 mm. A major part of chalcopyrite is enclosed in quartz grains (5–60 grains) with a size of 0.01–0.07 mm, and more seldomly, 0.06 mm × 0.17 mm. Chalcopyrite is sometimes located along the edges of quartz grains. The size of the mineral varies from 0.01 to 0.5 mm, and rarely up to 0.07–0.1 mm (Figure 5). Chalcopyrite on the edge of quartz grains occurs in the form of single grains and their clusters, sometimes in intergrowths with bornite. Moreover, there are quartz grains in which chalcopyrite adopts the form of fine impregnates with a size of 0.002–0.02 mm.

Chalcocite occurs more seldomly and primarily in the form of impregnations, in particular quartz grains (size 0.002–0.02 mm). The presence of fine-grained silica within particular quartz grains is characteristic (Figure 5). Separate chalcocite grains rarely occur, and usually contain from 1 to 4 grains (sizes 0.01–0.05, rarely up to 0.07 mm). In the vast majority of cases, chalcopyrite is enclosed in quartz grains (from scarce to 22 grains) with a size of 0.01–0.06 mm, and more seldomly up to 0.1 mm, intergrown with covellite and bornite, and the edge features quartz grains with a size of 0.01–0.06 mm, and more seldomly up to 0.1 mm (Figure 5). It is rarely encountered in intercalations with quartz (up to 0.15 mm) and around quartz grains (up to 0.05 mm × 0.2 mm).

Covellite has a secondary contribution to copper minerals (Figure 5E–H). It rarely occurs in free-form. As in the case of chalcopyrite and chalcocite, in the majority of cases, it

is enclosed in quartz grains with a size of 0.01–0.07 mm, and more seldomly up to 0.1 mm, or occurs in intercalations with chalcocite and bornite (size of up to 0.3 mm), and along the edges of grains with a size of 0.01–0.06 mm. Intercalations with chalcocite and bornite with a size of up to 0.1 mm are also recorded. Covellite rarely occurs in intercalations with quartz (up to 0.1 mm) and other copper minerals.

Like covellite, bornite is an important copper mineral (Figure 5B,D,E). It rarely occurs in free-form. In a vast majority of cases, like other copper minerals, it is enclosed within quartz grains with a size of 0.01–0.07 mm, together with chalcocite, covellite, and bornite, and along the edges of quartz grains. Intercalations of bornite with chalcocite, covelline, as well as bornite and chalcopyrite with a size of up to 0.1 mm, were observed. It rarely occurs with covelline in intercalations with quartz and other copper minerals. The majority of sulphates in OS are enclosed inside quartz grains in host rocks, and some of them occur individually.

### 5.2. Geochemical Characteristics

### 5.2.1. Geochemistry of Minerals by SEM-EDS

In the main ore minerals copper sulphides, and gangue quartz, K+feldspars, dolomite, clay minerals, and barite, the grains indicate oxidizing processes, with a relatively high content of oxide and low quantities of copper sulphides. Moreover, in ZM, barite was most common. The oxygen presence evidenced an oxidation process on the surface of samples. Silver and cobalt occur as an accessory and as additives, which are dispersed in the sediments (Figures 6 and 7).

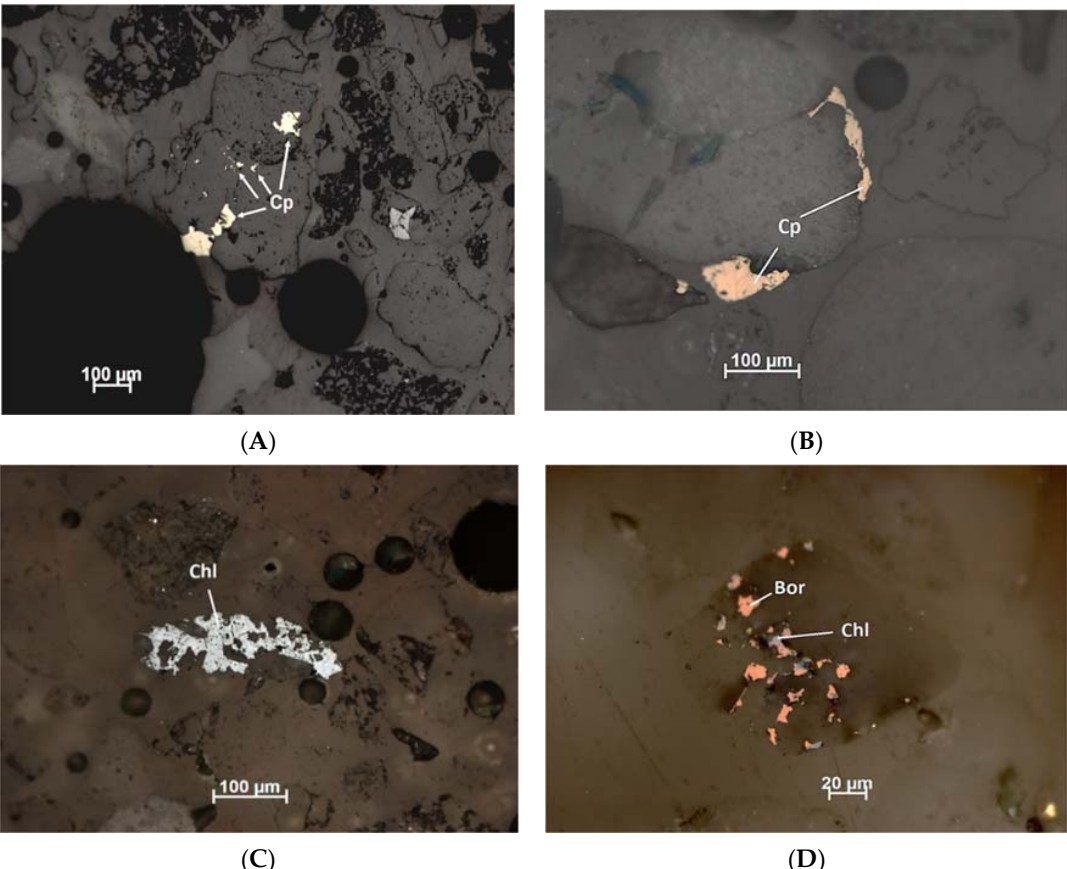

**Figure 5.** *Cont.*

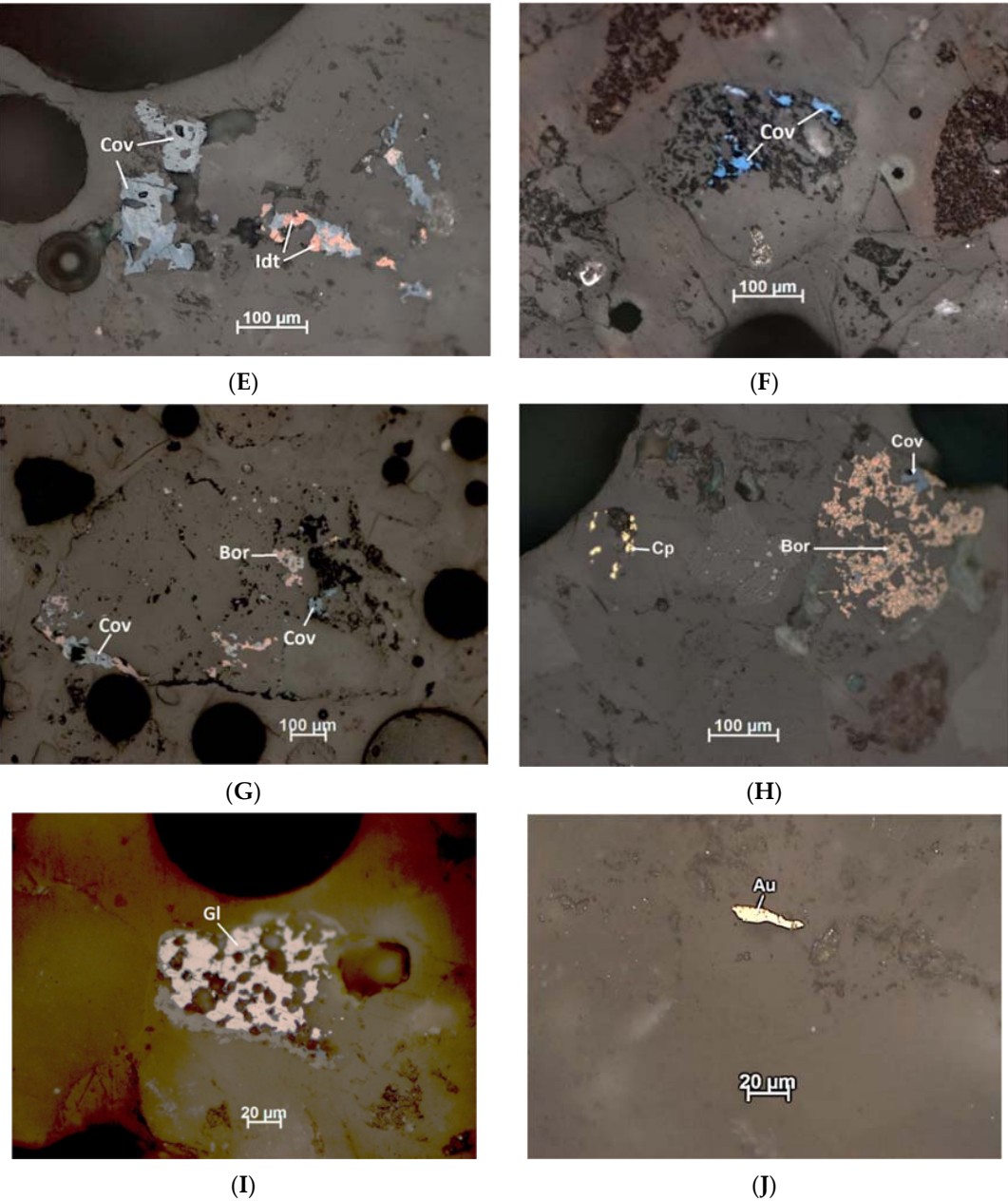

**Figure 5.** Examples of ore minerals from Osnovnoe sediments in microphotography, (**A**) chalcopyrite (Cp) grains in quartz; (**B**) chalcopyrite (Cp) along the edge of a quartz grain; (**C**) chalcocite (Cc) in intergrowth with quartz, 2/5; (**D**) Bornite (Bor) in intergrowth with chalcocite (Cc) in quartz; (**E**) covellite (Cov) in intergrowth with idaite (Idt); (**F**) covellite (Cov), Coveline (Cov) and Chalcopyrite along the edges of a quartz grain; (**G**) covellite (Cov) in intergrowth with bornite (Bor) in a quartz grain; (**H**) bornite (Bor)–covellite (Cov) intergrowths and chalcopyrite (Cp) in quartz grain (**I**) accumulation of galena (Gl) grains; (**J**) gold (Au) as native gold inclusion; 1 polarizer, reflected light.

### 5.2.2. Geochemistry of Deposits from ZM and OS

Dams in ZM show a similar chemical composition. In the northern dam, between 1230 and 2920 ppm Cu was found, and small variability of the chemical composition in vertical profiles was determined. The lowest values are observed in the deposits located closest to the pond, and the highest along the edge of the tailing pond, also characterised by the highest contribution of the silt and fine sand fractions (0.071–0.16 mm).

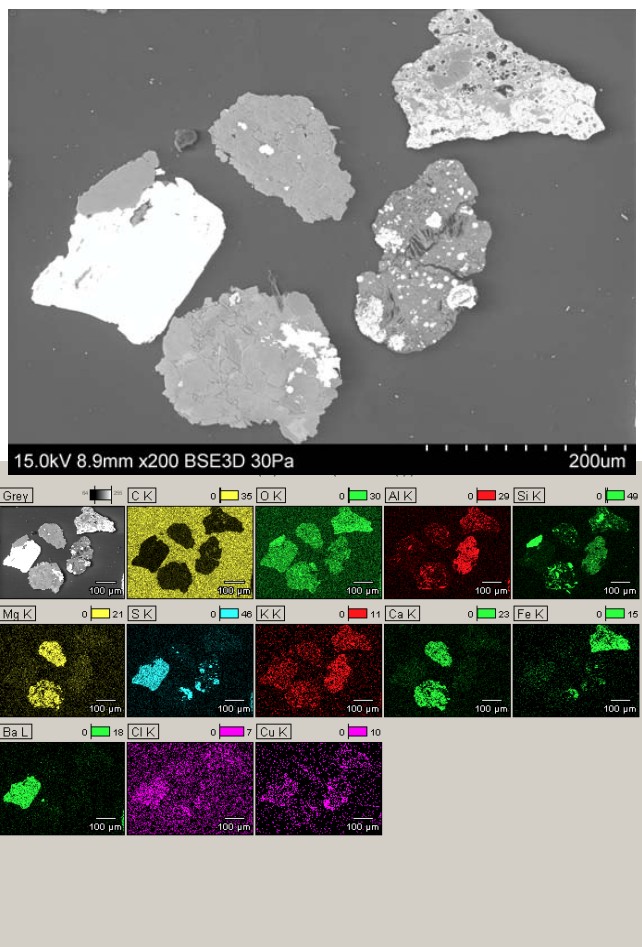

**Figure 6.** SEM–EDS analysis of heavy fraction of ZM. (upper part of Figure) BSE of sediments indicates mineralogical variable content; (lower part of Figure) metal diversity of the grains in the components of silica ($SiO_2$), K-feldspars, dolomite, clay minerals and barite, the grains indicate oxidizing processes, with a relatively high content of oxide and small quantities of copper sulphides.

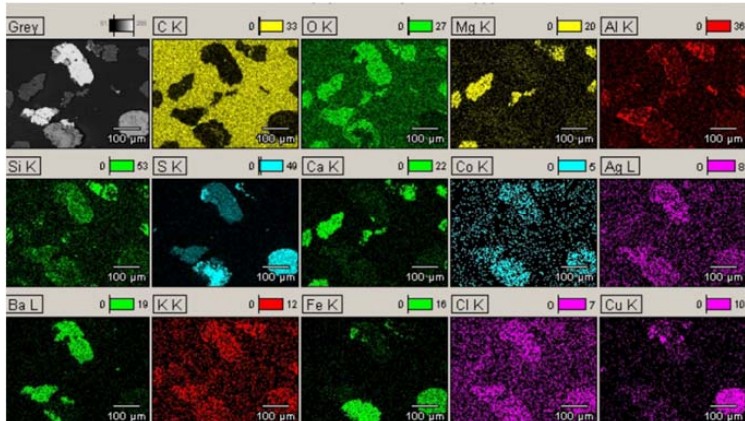

**Figure 7.** SEM–EDS analysis of heavy fraction of ZM. Representative SEM–EDS analysis, and distribution of chemical components in ore and gangue minerals: Fe-Cu-S indicate sulphides, Ba-S-O indicates sulphates, Si-Al-K indicates silicates (K feldspars), Mg-Ca indicates carbonates.

As in the copper content, the content of silver and arsenic is variable. In the 5 m profiles, the highest value (very high, hurricane value or sample), of up to 6250 ppm, evidently differs from the remaining ones. The mean copper content, determined in

34 samples from the western dam, is approximately 2290 ppm (approximately 0.2%). An increase in copper content in 5 m profiles occurs at a depth of 1 m and 1.5 m, and at a depth of 4.5 m below the surface of the tailing pond. An increase in depth is accompanied by a slight decrease in copper content. At the same depths, with copper content, the content of silver, molybdenum, and cobalt increases. Zinc and nickel show a more than double decrease in terms of depth content, and the contents of barium and strontium are variable. Copper content in ZM varies from 915 to 4440, but a mean Cu content in ZM of approximately 2280 ppm corresponds with the copper contents in OS and B. Lead content varies from 140 to 714 ppm in ZM-N (northern dam). In OS, the Pb concentration is somewhat higher, at approximately 1500 ppm. The zinc content varies from 29 ppm in ZM-N to 225 ppm in ZM-W (western dam) (Table S1), and in OS from 150 to 2500 ppm (Table S2). The content of Co, Cs, Ba, and Sr is somewhat higher in ZM than in OS, despite the presence of sulphates as an important component of SSC deposits in Poland. The semiquantitative spectral analysis of 35 samples in OS showed a copper content from 0.07–0.1%, 0.03–0.07% for lead, 0.01–0.03% for zinc, more than 1% for iron, and 0.5–0.7% for titanium (Table S2).

## 6. Statistical Correlation

Statistical correlation was performed based on the Pearson coefficient, calculated separately for samples from ZM and the tailing pond in OS.

The correlation matrix, based on ZM, shows a similar chemical composition. In the northern dam, between 1230 and 2920 ppm Cu was found, and small variability was determined in the chemical compositions of vertical profiles. The lowest values were observed in the deposits located closest to the pond, and the highest values were along the edge of the tailing pond, also characterised by the highest contribution of the silt and fine sand fractions (0.071–0.16 mm). As for the copper content, the contents of silver, nickel and molybdenum are variable (Table 1). The correlation matrix, based on Pearson coefficient calculations, shows mutual correlations in the distribution of elements in ZM, and points to groups of mutually correlated elements (Figure 8). A group of the following metals shows good positive correlation: Cu, Ag, Co, Ni, and Mo. Molybdenum also shows a weak correlation with vanadium and chromium. Lead is very evidently correlated with zinc (indicating a relatively high correlation coefficient). Barium and strontium are strongly mutually correlated. Next to barium, strontium is not correlated with metals with a group of rare earth elements. They are mutually correlated, and weakly correlated with V and Ni and elements forming lithoclasts. Relationships in the distribution of basic metal elements in OS (Figure 9) are similar to those in ZM, and different to ZM in terms of the distribution of trace and accompanying elements. A strong mutual correlation points to basic metals: Cu, Zn, and Pb, and Zn, with a stronger correlation to Cr and Mo than to Ni. The REE group shows a positive correlation with Ag and Cu, but no correlation or a negative correlation coefficient occurs for the other components (Figure 9).

**Table 1.** Minimum values, maximum values, averages, and median parameters for ZM and OS (on the basis of data from Tables S1 and S2 in Supplementary Materials).

| Parameter | Tailing | Cu | Pb | Zn | Ni | Co | Mo | Ag |
|---|---|---|---|---|---|---|---|---|
| Min. | ZM | 1410 | 259 | 46 | 8 | 8 | 5 | 4.1 |
| | OS | 700 | 250 | 150 | 30 | 20 | 0.3 | 0.2 |
| Max. | ZM | 6250 | 714 | 225 | 23 | 22 | 13 | 16 |
| | OS | 3000 | 1500 | 2500 | 70 | 50 | 2.5 | 0.5 |
| Aver. | ZM | 2389.1 | 365.9 | 93.4 | 10.7 | 11.3 | 8.3 | 5.9 |
| | OS | 1820.0 | 611.4 | 762.9 | 40.9 | 28.0 | 1.5 | 0.3 |
| Median | ZM | 2110 | 354 | 81 | 10 | 11 | 8 | 53 |
| | OS | 1500 | 500 | 700 | 30 | 30 | 1.5 | 0.30 |

| ZM | Cu | Pb | Zn | Ni | Co | Mo | Ag | V | Cr | Ba | Sr | Y | Yb | La | Ce |
|---|---|---|---|---|---|---|---|---|---|---|---|---|---|---|---|
| Cu | 1 | | | | | | | | | | | | | | |
| Pb | 0.547 | 1 | | | | | | | | | | | | | |
| Zn | 0.645 | 0.826 | 1 | | | | | | | | | | | | |
| Ni | 0.645 | 0.526 | 0.342 | 1 | | | | | | | | | | | |
| Co | 0.116 | 0.110 | 0.083 | 0.502 | 1 | | | | | | | | | | |
| Mo | 0.536 | 0.611 | 0.502 | 0.641 | 0.193 | 1 | | | | | | | | | |
| Ag | 0.596 | 0.620 | 0.503 | 0.662 | 0.321 | 0.515 | 1 | | | | | | | | |
| V | 0.084 | 0.259 | 0.092 | 0.713 | 0.359 | 0.581 | 0.344 | 1 | | | | | | | |
| Cr | −0.013 | 0.109 | −0.089 | 0.338 | 0.050 | 0.281 | 0.142 | 0.284 | 1 | | | | | | |
| Ba | −0.090 | −0.312 | −0.170 | −0.383 | −0.198 | −0.275 | −0.357 | −0.223 | −0.213 | 1 | | | | | |
| Sr | 0.060 | −0.213 | 0.002 | −0.311 | −0.276 | 0.008 | −0.461 | −0.008 | −0.156 | 0.644 | 1 | | | | |
| Y | −0.237 | 0.092 | −0.155 | 0.496 | 0.139 | 0.263 | 0.163 | 0.656 | 0.166 | −0.013 | −0.089 | 1 | | | |
| Yb | −0.243 | 0.106 | −0.124 | 0.438 | 0.343 | 0.151 | 0.286 | 0.521 | 0.199 | 0.019 | −0.205 | 0.612 | 1 | | |
| La | −0.173 | 0.100 | −0.118 | 0.414 | 0.141 | 0.263 | 0.169 | 0.694 | 0.248 | −0.076 | −0.041 | 0.603 | 0.664 | 1 | |
| Ce | −0.395 | −0.035 | −0.190 | 0.297 | 0.183 | −0.006 | 0.189 | 0.432 | 0.199 | −0.124 | −0.246 | 0.483 | 0.675 | 0.705 | 1 |

**Figure 8.** Correlation coefficient of chemical components in ZM. The highest Pearson coefficient indicate pairs: Zn–Pb, Zn–Pb, Ni–Mo, Ag and V, Ba–Sr, La-Ce–Yb.

| OS | Cu | Pb | Zn | Ni | Co | Mo | Ag | V | Cr | Ba | Sr | Y | Yb | La | Ce |
|---|---|---|---|---|---|---|---|---|---|---|---|---|---|---|---|
| Cu | 1 | | | | | | | | | | | | | | |
| Pb | 0.694 | 1 | | | | | | | | | | | | | |
| Zn | 0.477 | 0.763 | 1 | | | | | | | | | | | | |
| Ni | 0.234 | 0.207 | 0.253 | 1 | | | | | | | | | | | |
| Co | 0.274 | 0.555 | 0.556 | 0.383 | 1 | | | | | | | | | | |
| Mo | 0.291 | 0.284 | 0.630 | 0.531 | 0.518 | 1 | | | | | | | | | |
| Ag | 0.523 | 0.399 | 0.178 | -0.075 | 0.323 | 0.020 | 1 | | | | | | | | |
| V | 0.311 | 0.213 | 0.3157 | 0.645 | 0.568 | 0.635 | 0.031 | 1 | | | | | | | |
| Cr | 0.417 | 0.552 | 0.641 | 0.391 | 0.746 | 0.583 | 0.299 | 0.437 | 1 | | | | | | |
| Ba | −0.149 | −0.065 | 0.293 | −0.146 | 0.171 | 0.305 | −0.176 | −0.095 | 0.237 | 1 | | | | | |
| Sr | 0.063 | 0.108 | 0.101 | 0.388 | 0.388 | 0.318 | 0.199 | 0.300 | 0.560 | 0.010 | 1 | | | | |
| Y | 0.026 | −0.191 | −0.152 | 0.090 | −0.222 | 0.188 | 0.102 | 0.146 | 0.097 | −0.338 | 0.247 | 1 | | | |
| Yb | 0.097 | 0.215 | 0.049 | −0.101 | 0.014 | 0.006 | 0.418 | −0.060 | 0.016 | −0.292 | 0.248 | 0.429 | 1 | | |
| La | 0.149 | −0.154 | −0.387 | −0.384 | −0.579 | −0.309 | 0.242 | −0.434 | −0.409 | −0.305 | −0.135 | 0.265 | 0.183 | 1 | |
| Ce | 0.359 | 0.105 | −0.114 | −0.193 | −0.413 | −0.114 | 0.245 | −0.195 | −0.355 | −0.227 | −0.135 | 0.078 | 0.224 | 0.760 | 1 |

**Figure 9.** Correlation coefficient of chemical components in OS. The highest Pearson coefficient indicate the following pairs: Zn–Pb, Cu–Pb, Zn–Mo, Zn–Cr, Ni–V, Ag and V, La–Ce, Sr–Cr, Co, Ni, Mo.

## 7. Discussion

Sediments from tailings in ZM and OS are similar according to grain size and mineralogical components; however, due to chemistry, more differences are observed.

Grain size changes from fine sands at the crown of the sedimentation tank to silts at the tailing pond. The grain size of material from ZM, B, and OS is primarily determined by the degree of ore fragmentation obtained in the processing plant (to a fraction of 0.074 mm). The means of material deposition determines the lateral variability in grain size distribution (coarse deposits accumulate at the crown of the dams, fine deposits accumulate in ponds). This pattern also concerns other tailings in Poland [41,44] and around the globe [45]. In terms of mineral composition, deposits in the tailing pond reflect the ores occurring in the deposit—they are largely composed of the same ore components (copper sulphides, copper and iron sulphides)—and rocks: fragments of dolomites, sandstones, and shales, single grains of quartz and feldspars in ZM, and sandstones and quartz grains in B and OS (Figures 1–4). In tailings from OS ore minerals, smaller sizes are indicated than those in ZM. The small grains facilitated the chemical treatment and metal extraction [20,32] in coarser fractions from the crown of construction, which enable mechanical separation in first step of enrichment. Chalcopyrite grains are the most frequently occurring component in separate grains in both ZM and OS tailings (Figures 4 and 5), useful for copper extraction [32,46–48].

Mean copper content is approximately 0.2 wt% (maximum copper content increased to 3000 ppm (on average, 1820 ppm), and its content is also provided by older sources, such as technogenic formations [49,50]. In tailings of the Zhezgazgan OS deposit, the amount does not deviate from the original composition of the waste, which results from processing. Copper content in waste is related to copper recovery in concentrates, averaging at 88%.

The remaining portion is supplied to tailing waste. Copper content in ZM varied from 915 to 4440, but the mean Cu content in ZM of approximately 2280 ppm is in accordance with copper content in OS. Lead content varies from 140 to 714 ppm in ZM-N (northern dam). In OS, Pb concentration is somewhat higher, reaching approximately 1500 ppm. Zinc content varies from 29 ppm in ZM-N to 225 ppm in ZM-W (western dam), and in OS from 150 to 2500 ppm (Table 1, Tables S1 and S2). The content of Co, Cs, Ba, and Sr is somewhat higher in ZM than in OS, despite the presence of Ba and Sr sulphates as important components of SSC deposits in Poland [26,31,45]. Silver, as one of the primary metals recovered in Poland, shows high concentrations in ZM. Silver content varies from 3 to 16 ppm, molybdenum from 3 to 13 ppm, and Co from 6 to 22 ppm. In Zhezkazgan, silver content varied from 0.2 to 0.3 ppm, molybdenum from 0.3 to 0.5 ppm, and Co content varied from 20 to 50 ppm. The higher contents of Co and Pb in OS compared to ZM correspond with higher concentrations of these metals in the primary ores, or a different processing pattern in deposits of Zhezkazgan and LGOM.

In both tailings, ZM and OS, copper is primarily accumulated in the form of Cu-Fe-S or Cu-S sulphates. Metals accompanying Ag, Zn, Pb, and Ni, correlated with copper distribution, occur in the form of sulphides, and considerably more seldomly in ZM than in the oxidised carbonate forms of malachite and cerussite (Figure 4h). The positive correlation between copper and the accompanying metals, Pb, Zn, Ni, Co, Ni, and Mo, evidenced in both ZM and OS, permits the collective extraction of the metal from ore minerals. In OS and B tailings from Kazakhstan, the metals are related to ore minerals in sandstone (Figure 5A–J).

Pearson correlations show that the primary components of SSC deposits mutually correlate well in both ZM and OS. Lead is strongly correlated with zinc and weakly with copper, occurring in the form of galena or cerussite. The Pb-Zn correlation points to the mutual association of sphalerite and galena. This is a pattern observed in all SSC deposits. In ZM, silver is correlated most strongly with copper, and Ag is also positively correlated with Mo and Ni (Figures 8 and 9). Although the analysed deposits showed no presence of silver sulphides by means of the microscopic method, admixtures of silver were confirmed in SEM–EDS research under an electron microscope (Figure 6). Microchemical research showed the contribution of silver in the structures of Cu-S and Cu-Fe-S minerals, particularly covellite and bornite [41]. Moreover, Cu, Zn, Pb, and Ag, and Au in deposits from OS were described in waste-rock-forming minerals (quartz and feldspars) (Figure 5) [43].

In ZM, an increase in the grains and sandstone in a fraction from 0.025 to 0.5 is accompanied by an increase in Cu, Ag, Zn, Ni, and Pb content. Therefore, more copper occurs in thicker deposits, forming the tailing dams. Ni, Co, and Mo are correlated with two groups of elements: the base metals Cu, Zn, and Pb (correlation coefficient higher than 0.5), and rare earth elements (REE), related to fine grain fractions. The concentration of copper and other base metals in ZM increases with an increase in coarse fractions in the deposits (more than 0.071 mm). Grain size structure in tailing deposits is, therefore, important, because it causes higher concentrations of copper and correlated metals. Copper concentration changes cyclically, depending on changes in the contribution of the coarse- and fine-grained fractions, resulting from a cyclical supply of material to the tailing pond [41,43] and, possibly, from oxidizing processes [31,51].

Vanadium and chromium in ZM are positively correlated with REE, whereas in the OS, the correlation with REE is negative. An increase in REE concentration is related to the presence of black shales; therefore, these metals are related to this type of technogenic ore. In ZM, Ni, Co, and Cr, a negative correlation with REE and the occurrence of Cr and V as separate phases from LREE in the Zhezgazgan region were confirmed. Barium and strontium are strongly mutually correlated with ZM, and weakly correlated with OS. Barium, as a highly mobile element in ZM, is related to the presence of their minerals, barite and strontianite (Figures 6 and 7), and is related to Zechstein salt cyclothem rocks in SSC deposits in LGOM [52,53]. A weak correlation between Ba and Sr in OS is related

to their high mobility and their low concentrations, due to the lack of their own minerals. Moreover, in the OS, Sr is weakly positively correlated with metals from the HREE group, pointing to common mineral phases from the group of heavy rare-earth elements in the Carboniferous SSC deposits in Kazakhstan (Figure 9).

The mineral composition of tailings is determined by the type of ore: in the Zhezkazgan deposit, they originate from copper-bearing sandstone, and in the LGOM, deposits from sandstone, dolomite, and copper-bearing shales. A small admixture of copper-bearing shales in the northern and western dam of the ZM tailing pond suggests the high efficiency of the floatation processes, and the recovery of metals from useful minerals in the richest ore in a Kupferschiefer type deposit. Moreover, all the tailing deposits showed a negligible contribution of chalcocite, the basic mineral in SSC deposits, towards the primary ores, which also points to the high efficiency of the floatation process, permitting copper recovery from the most important copper-bearing mineral in both deposits [5,6,12,54,55]. The volume of the accumulated deposits in both tailing deposits from B and OS in Kazakhstan exceeds 900.00 million tonnes, or 617.00 million m$^3$. The amount of base metal in tailings reaching 1190 thousand t Cu (at a mean Cu content of 0.128%), Pb = 175 thousand t (0.02%), Zn = 250 thousand t (0.03%), and Ag = 2100 t (2.46 g/t) is sufficiently high to allow for the recovery of metals from the tailing ponds. Polish law does not allow for the exploitation of metals from tailing ponds after copper floatation at present. The storage of copper floatation deposits in post-mining shafts is planned in the newly discovered deposits in the vicinity of Żory, and newly discovered deposits in Kamoa, Kakula [10,56]. The new copper discovery and new exploitation project correspond to the high copper market demands [18,19,51,52,57].

Tailings in Zhezkazgan accumulated ore, which could be processed as a technogenic deposit of metals with valuable related elements. Tailing-processing technologies should ensure the recovery of ore minerals enclosed in siliceous grains to improve the chance of metal recovery from tailings and the rational exploitation of the accumulated components. It should be noted that the enclosed sulphides are related to rock grains smaller than 100 μm, are easily available for leaching, and an increase in the completeness of metal extraction from sulphides inside rock grains larger than 150–200 μm requires additional measures during cyanidation. Notably, these processes cannot be provided at present, and metal resources could be treated as having economical potential.

## 8. Conclusions

Technogenic deposits from tailings from two different SSC deposits showed a mineralogical trace similar to that of primary ores. Waste components in deposits from Kazakhstan included fragments of sandstone and single quartz grains, and this mineralogical composition, as well as the concentration in fine fractions, creates opportunities for potential economic interests. The mineralogy of sediments from tailing in Poland is more complex and more differentiated, according to their granulometric characteristics. The sediments consist of fragments of sandstones, dolomites, and copper-bearing shales, with copper, zinc and lead sulphide additives forming both intergrowths and individual, separate grains, which could be more problematic, considering their potential economic recovery. The composition and quantity of metals and ore minerals in tailings documented the relevant mineral processing and appropriate yield of metals through floatation processes in both copper districts.

Tailings from two deposits were found to contain the same group of the following ore minerals: chalcopyrite; covellite, digenite, chalcocite, and bornite. Other minerals include galena, sphalerite, pyrite, and native gold in the Zhezkazgan deposits. Base metals Cu, Pb, Zn, and the accompanying cobalt and silver, occur in small amounts. Statistically, a good mutual correlation of the base metals Cu, Zn, and Pb results from the main ore minerals, constituting carriers of metals which are typical of SSC deposits.

One of the sources of development of the raw material base of processing plants are tailings, which can be considered potential technogenic deposits. Cu, Pb, Zn, Ag, Mo, and

Co show concentrations that can offer a perspective for recovery in tailings. Although they are not achieving the technologic parameters demanded for the exploitation of copper ore at present, they should be taken into consideration as being potentially interesting for future metallurgy.

**Supplementary Materials:** The following are available online at https://www.mdpi.com/article/10.3390/min11070752/s1: Ore Minerals and Metal Distribution in Tailings of Sediment-hosted Stratiform Copper Deposits from Poland and Kazakhstan; Table S1. Ore Minerals and Metals Distribution Table 1. Table S2. Ore Minerals and Metals Distribution Table 2.

**Author Contributions:** Conceptualization, A.D.-C. and A.B.(Adilkhan Baibatsha); methodology, A.D.-C., A.B. (Adilkhan Baibatsha) and A.B. (Alma Bekbotayeva); validation, A.B. (Adilkhan Baibatsha) and A.B. (Alma Bekbotayeva); formal analysis, A.D.-C. and A.B. (Adilkhan Baibatsha); investigation, A.D.-C., A.B. (Adilkhan Baibatsha), A.B. (Alma Bekbotayeva) G.O. and A.B. (Akmaral Baisalova); writing—original draft preparation, A.D.-C. and A.B. (Adilkhan Baibatsha), A.B. (Alma Bekbotayeva) and G.O.; writing—review and editing, A.D.-C. and A.B. (Adilkhan Baibatsha); visualization A.D.-C. and A.B. (Adilkhan Baibatsha). All authors have read and agreed to the published version of the manuscript.

**Funding:** This research was supported by the scientific program BR05233713: comprehensive geological study of subsurface resources for the development of resource base and mining exploitation of new sources of ore raw materials in Kazakhstan, granted to A. B. Baibatsha and Grant no. NN307106035 Ministry of Science and Higher Education, on the subject: mineralogy and geochemistry of sediments in the northern and western dam of Żelazny Most, granted to A. Duczmal-Czernikiewicz.

**Institutional Review Board Statement:** Not applicable.

**Informed Consent Statement:** Not applicable.

**Data Availability Statement:** No data availability was provided with the paper.

**Acknowledgments:** A.D.-C. expresses his gratitude to the Management and Employees of Zakład Hydrotechniczny KGHM Polska Miedź SA for allowing the sampling, and Katarzyna Zielnica, and Pawel Faryniarz, MSC, graduate in Institute Geology, for help in the field research. The authors are grateful to the two anonymous reviewers for valuable comments that improved the quality of the manuscript.

**Conflicts of Interest:** The authors declare no conflict of interest.

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
