# Peer review of "Ore Minerals and Metal Distribution in Tailings of Sediment-Hosted Stratiform Copper Deposits from Poland and Kazakhstan"

_minerals, doi:10.3390/min11070752_

Round 1
Reviewer 1 Report
English needs addressing throughout. As does spelling.
Figure 2 -image d missing and image c apparently mislabeled.
Problem throughout paper confusing sulphates with sulphides
In general it appears sulfides in Zhezkazgan are encased in quartz – higher grind needed to liberate. In Poland unclear – dominant sulfide grains but more mineralogical variation – need to work on floatation to increase sulfide yield?
Section 5.2.1 especially requires rewriting.
Discussion overly long and repetitive with paper – could be reduced significantly and would improve the paper.
Data actually suggests none of the tailings material may be suitable as technogenic “ore” – Ore implies that the material is currently economic to produce metal from. Looking at data presented I am pretty sure that none of it is suitable for economic production of metal. The metallurgists and mining engineers in both countries did a very good job or extracting maximum metal from what was mined on the first pass.
Author Response
Response to Reviewer 1 Comments
Point 1: English needs addressing throughout. As does spelling.

Response 1: English have been corrected by the native English.
Point 2: Figure 2 -image d missing and image c apparently mislabeled.
Response 2: Figure 2 d image have been put again and c – has new label.
Point 3: Problem throughout paper confusing sulphates with sulphides
Response 3: The sulphates and sulphides have been corrected in the text.
Point 4: In general it appears sulfides in Zhezkazgan are encased in quartz – higher grind needed to liberate. In Poland unclear – dominant sulfide grains but more mineralogical variation – need to work on floatation to increase sulfide yield?
Response 4: The sulfides in Zhezkazgan are enclosed in quartz and higher grind needed to liberate. In Poland there are three types of ore – sandstones, shales and carbonates, and inside the gangue minerals there are sulfides as intergrowths. Moreover there are a lot of individual grains of sulphides. Indeed the floatation could help to increase of sulphide yield in Poland. This part was put in last chapter.
Point 5. Section 5.2.1 especially requires rewriting
Response 5: Section 5.2.1. have been rewritten, from lines 378 to the 384, missing letters and wrong corrected.
Point 6. Discussion overly long and repetitive with paper – could be reduced significantly and would improve the paper.
Response 6: Discussion have been shortened and reduced significantly, according to suggestion.
Point 7. Data actually suggests none of the tailings material may be suitable as technogenic “ore” – Ore implies that the material is currently economic to produce metal from. Looking at data presented I am pretty sure that none of it is suitable for economic production of metal. The metallurgists and mining engineers in both countries did a very good job or extracting maximum metal from what was mined on the first pass
Response 7: The term “technogenic ore” have been changed into the wastes after flotation, and were described as technogenic sediments or the remnants after flotation process. We agreed, that the wastes studied cannot be treated as copper ore at that moment, because they are not as much copper content as really copper ore, what was clarified and included in discussion and conclusions.
Thank you for the valuable comments

Reviewer 2 Report
Dear Authors,
Your manuscript deals with the Ore Minerals and Metal Distribution in Tailings of Sediment-hosted Stratiform Copper Deposits from Poland and Kazakhstan, presenting a combination of a large amount of data. In my opinion is an effort worthy of publication at this journal after some moderate to major revision. Revision is concerning with some rearrangements and other specific comments:
Abstract: Consider rewriting
Line 36-37 : provide refs
Line 38: …many Sediment-hosted Stratiform Copper (SSC)…
Line 58: …of hazardous components. Such as?????
Line 59: how can the impermanent character of the dam be a hazard?
Line 63: what is KGHM Polska Miedz SA???
Line 70 … and several million…. How many??? 10? 100?
Line 82: Start with The subject of this study
Lines 94-96: add refs
Line 97: … of this type…
Lines 100-102: consider rewriting. The repetition is obvious
Line 108: how considerably ???
Lines 186,187:Did you presented pictures under transmitted light? How the thin sections have been coated for SEM analysis, please add
Chapter 2 could have been a part of the Introduction
Chapter 2.1: Where is Lower Silesia?? Where is the location of the old copper belt?? Introduce a relevant map and a stratigraphic column and rewrite the paragraph accordingly.
Chapters 2.2, 3.1, 3.2: do as Chapter 2.1
Use commas for thousands and millions
Figure 1: add explanatory legends in d and e. Add scale bars to a and c
Figure 2: d is missing. The figure caption is confusing
Figure 3: add under reflected light
Lines 237-238: rewrite or delete them
Lines 318-319: How did you calculate the mineral amounts , please add in section Materials and Methods
Line 361: correct the word minerals
Line 420: Table A1 or Si, please check
Line 457: give at least 2 more refs
Conclusion: you are practically repeating the Discussion. It should be either more to the point or even better merged with discussion under the title Discussion and Conclusions. Try to provide specific estimation as results since you have all the required data.
Kind regards
Author Response
Response to Reviewer 2 Comments
Point 1: Abstract: Consider rewriting
Response 1: Abstract have ben rewritten and corrected
Point 2: . Line 36-37 : provide refs
Response 2: refs have been provided in the text and Extended bibliography
Point 3: Line 38: …many Sediment-hosted Stratiform Copper (SSC)…
Response 3: the text was changed in line 38, according to suggestion there were changed to “many sediment-hosted stratiform copper (SSC), instead of SSC
Point 4: Line 58: ……of hazardous components. Such as?????
Response 4: there were put in the text text - line 58,such as As and Cd
Point 5: how can the impermanent character of the dam be a hazard?
Response 5: The dam is formed very careful and it is permanent although in the past the construction of the dam have been ceased and disrupting were happen, so the dam could be dangerous to the environment and not safe to society. The text have been changed according to suggestion. Thank you.
Point 6: Line 63: what is KGHM Polska Miedz SA???
Response 6: Line 63: English translation have been used : KGHM (shorthand from polish name for Leader of copper and silver production of Polish Copper District; in English: the Polish State Mining and Metallurgical Combine.
Point 7: Line 70 … and several million…. How many??? 10? 100?
Response 7: Line 70: and about 1 million tons of sediments deposited in all tailings in Poland .
Point 8: Line 82: Start with The subject of this study
Response 8: Line 82: done
Point 9: Lines 94-96: add refs
Response 9: the references have been added [22,23]
Point 10: Line 97: … of this type…
Response 10: “ the “ has been changed on “this type”
Point 11: Lines 100-102: consider rewriting. The repetition is obvious
Response 11: the line 100-101 repetition in the text have been omitted.
Point 12: Line 108: how considerably ???
Response 12: The text have been extended: and at the moment the copper is not produced in the Zehzkazgan deposits.
Point 13: Lines 186,187: Did you presented pictures under transmitted light? How the thin sections have been coated for SEM analysis, please add
Response 13: the SEM EDS have not been coated; the explanation has been add in the text
Point 14: Chapter 2 could have been a part of the Introduction
Response 14: Chapter 2 have been transferred into the Introduction: Lines 55-71.
Point 15: Chapter 2.1: Where is Lower Silesia?? Where is the location of the old copper belt?? Introduce a relevant map and a stratigraphic column and rewrite the paragraph accordingly.
Response 15: The map of deposits in Poland have been attached in the text as new Figure 1.
Point 16: Chapters 2.2, 3.1, 3.2: do as Chapter 2.1
Response 16: Chapter 2.1, and Chapters 2.2, 3.1, 3.2: are on the new Figure 2. Former figures have been re-numbered.
Point 17: Use commas for thousands and millions
Response 17: commas for thousands and millions have been used in the chapters 2 and 3
Point 18. Figure 1: add explanatory legends in d and e. Add scale bars to a and c
Response 18: the former Figure 1 is labeled in this version as Figure 2. The hammer as a scale bars on the Figure 1B
Point 19. Figure 2: d is missing. The figure caption is confusing
Response 19: Figure 2 (in new version Figure 3 was changed, the caption was corrected according to Reviewer suggestion
Point 20. Figure 3: add under reflected light
Response 20: in the legends in former Figures 3 and 4 the reflected light was added, instead of opaque light
Point 21. Lines 237-238: rewrite or delete them
Response 21: Lines 237-238 were deleted
Point 22. Lines 318-319: How did you calculate the mineral amounts , please add in section Materials and Methods
Response 22: In Lines 318-319; following text was add in the section Materials and Methods
Quantitative calculations under a microscope were performed with a conventional eyepiece, along with the determination of the optical properties of minerals. To avoid repeated counts in one field of view, the movements of the thin section were required.
The essence of the calculation is to assess that part of the field of view, which is occupied by a given mineral. After calculating each mineral over the area of the entire thin section, the quantitative ratios of the minerals in it were calculated.
The cross of the eyepiece divides the field of view into four equal parts. The percentage can be calculated relative to the entire field of view, given that every quarter is 25%. For example, if a mineral occupies approximately half a quarter of the field of view, then its content in this case amount 12.5%, and if the fourth part - 6%. The quantitative ratios obtained in one field of view were recorded and in a similar way, by moving the thin section the content of minerals in the remaining parts was calculated over the entire area of the section. The arithmetical mean was calculated using particular data.
Point 23. Line 361: correct the word minerals
Response 23: the text in 361 line was changed and wrong letters corrected
Point 24 Line 420: Table A1 or Si, please check
Response 24: Line 420 was changed: the S1 instead A1 in the text
Point 25 Line 457: give at least 2 more refs
Response 25: 3 additional references are listed in literature list
Point 26 Conclusion: you are practically repeating the Discussion. It should be either more to the point or even better merged with discussion under the title Discussion and Conclusions. Try to provide specific estimation as results since you have all the required data.
Response 26: The conclusions were rewritten and shortened, we hope in good way.
Thank you very much for the valuable and helpful comments.

Round 2
Reviewer 2 Report
Please explain abbreviations in abstract